# A Nomogram Based on Preoperative Lipiodol Deposition after Sequential Retreatment with Transarterial Chemoembolization to Predict Prognoses for Intermediate-Stage Hepatocellular Carcinoma

**DOI:** 10.3390/jpm12091375

**Published:** 2022-08-25

**Authors:** Xiang-Ke Niu, Xiao-Feng He

**Affiliations:** 1Department of Interventional Radiology, Affiliated Hospital of Chengdu University, Chengdu 610081, China; 2Department of Interventional Radiology, Nanfang Hospital, Southern Medical University, Guangzhou 510515, China

**Keywords:** transarterial chemoembolization, hepatocellular carcinoma, lipiodol, nomogram, overall survival

## Abstract

(1) Background: Conventional transarterial chemoembolization (cTACE) is the mainstay treatment for patients with Barcelona Clinic Liver Cancer (BCLC) B-stage hepatocellular carcinoma (HCC). However, BCLC B-stage patients treated with cTACE represent a prognostically heterogeneous population. We aim to develop and validate a lipiodol-deposition-based nomogram for predicting the long-term survival of BCLC B-stage HCC patients after sequential cTACE. (2) Methods: In this retrospective study, 229 intermediate-stage HCC patients from two hospitals were separately allocated to a training cohort (n = 142) and a validation cohort (n = 87); these patients underwent repeated TACE (≥4 TACE sessions) between May 2010 and May 2017. Lipiodol deposition was assessed by semiautomatic volumetric measurement with multidetector computed tomography (MDCT) before cTACE and was characterized by two ordinal levels: ≤50% (low) and >50% (high). A clinical lipiodol deposition nomogram was constructed based on independent risk factors identified by univariate and multivariate Cox regression analyses, and the optimal cutoff points were obtained. Prediction models were assessed by time-dependent receiver-operating characteristic curves, calibration curves, and decision curve analysis. (3) Results: The median number of TACE sessions was five (range, 4–7) in both cohorts. Before the TACE-3 sessions, the newly constructed nomogram based on lipiodol deposition achieved desirable diagnostic performance in the training and validation cohorts with AUCs of 0.72 (95% CI, 0.69–0.74) and 0.71 (95% CI, 0.68–0.73), respectively, and demonstrated higher predictive ability compared with previously published prognostic models (all *p* < 0.05). The prognostic nomogram obtained good clinical usefulness in predicting the patient outcomes after TACE. (4) Conclusions: Based on each pre-TACE lipiodol deposition, two sessions are recommended before abandoning cTACE or combining treatment for patients with intermediate-stage HCC. Furthermore, the nomogram based on pre-TACE-3 lipiodol deposition can be used to predict the prognoses of patients with BCLC B-stage HCC.

## 1. Introduction

Hepatocellular carcinoma (HCC) is the fifth most commonly diagnosed form of cancer and the second leading cause of cancer-related death worldwide [1]. Patients with Barcelona Clinic Liver Cancer (BCLC) stage B disease are considered ineligible for surgical resection, and the median overall survival (OS) is approximately 2 years, even after optimal treatment [2]. Conventional transarterial chemoembolization (cTACE) is one of the most widely performed digital subtraction angiography (DSA)-guided catheter-based therapies for the treatment of BCLC stage B HCC, and a meta-analysis based on randomized controlled trials showed that cTACE has a positive effect on survival in patients in this stage of disease [3]. cTACE generally uses ethiodized oil, or lipiodol, mixed with chemotherapeutic agents. This mixed liquid is superselectively injected into tumor-feeding arteries, followed by bland embolization of the tumor blood supply [4].

However, some questions remain to be addressed. First, resulting from the enormous heterogeneity of BCLC stage B HCC patients, the prediction of outcome is also heterogeneous for patients treated with cTACE [5,6]. Various scoring systems predicting the prognosis of HCC patients receiving cTACE are available, such as albumin-bilirubin (ABLI) grade and up-to-7 criteria [5,7]. Unfortunately, these predictive models need sophisticated calculation and are not fully validated. Second, a previous study by Hiraoka et al. [8] concluded that repeated TACE gradually reduces hepatic reserve function. Moreover, in clinical practice, some patients cannot tolerate TACE treatment, which may manifest as TACE resistance. Thus, an appropriate judgment of the number of TACEs that should be performed has become important to avoid harmful TACE and for less-effective patients switching to multiple systemic therapies in a timely and effective manner.

Lipiodol, an injectable agent, can be visualized with pretreatment multidetector computed tomography (MDCT). Several studies have shown that lipiodol can be detected within a treated tumor for several months after injection [9,10,11]. Lipiodol deposition on intraprocedural cone-beam computed tomography (CBCT) has been utilized to predict treatment response [12,13,14]. Unfortunately, CBCT is not widely used, especially in developing countries. In routine clinical practice, MDCT is applied to guide treatment planning before each TACE procedure. In this study, based on a 3D quantification of preoperative lipiodol deposition, we develop and validate a new predictive nomogram for assessing the prognoses of patients with BCLC B-stage HCC after sequential cTACE treatment, which may be used to assess individualized prognosis and can help to select patients suitable for sequential cTACE treatments.

## 2. Methods

### 2.1. Patients and Tumor Selection

The diagnosis of HCC was based on pathology or noninvasive imaging features outlined by the American Association for the Study of Liver Disease guidelines [15,16]. Patients treated with TACE at Southern Medical University Nanfang Hospital from May 2010 to May 2017 were included in the training cohort. From May 2010 to May 2017, the independent validation cohort consisted of patients who underwent TACE treatments at the Affiliated Hospital of Chengdu University. The study included patients who were partially reported by our recent research [17]. We used the following inclusion criteria in both cohorts (Figure 1): (a) BCLC B-stage disease with preserved liver function (Child–Pugh class A or B); (b) patients were ≥ 18 years old with a performance status (PS) score ≤ 2 at the time of the first TACE treatment; (c) preoperative MDCT imaging was performed within 24 h–72 h prior to each TACE treatment; and (d) TACE was performed as monotherapy, and at least four TACE sessions were performed by a single patient. Overall survival (OS) was defined as the interval from the time of each cTACE session to the time of death or last follow-up. This study was censored on 15 March 2020.

The study was approved by the Ethics Committee of the Affiliated Hospital of Chengdu University and the Ethics Committee of the Southern Medical University Nanfang Hospital. The study protocol conformed to the ethical guidelines of the 1975 Declaration of Helsinki. All patients or their relatives provided written informed consent.

### 2.2. cTACE Protocol

Briefly, we performed superselective catheterization of the tumor-feeding branches with a microcatheter, and an emulsion containing a 50 mg doxorubicin (Adriamycin; Pharmacia & Upjohn, Peapack, NJ, USA) mixture with lipiodol (Lipiodol; Guerbet, Paris, France) was infused with a microcatheter, followed by delivery of a gelfoam slurry (Upjohn, Kalamazoo, MI, USA) or microsphere particles (Embosphere Microspheres; Biosphere Medical, Rockland, MA, USA) until tumor blood flow stagnation was seen on DSA imaging.

cTACE was performed by interventional radiologists with 10 years of experience in hepatic interventions at each institution. The procedure was conducted on demand, and cTACE was discontinued in the case of a complete radiological response. In addition, the presence of Child–Pugh type C cirrhosis, vascular invasion, extensive liver involvement, extrahepatic metastases, or a PS score >2 were considered contraindications to TACE retreatment [18].

### 2.3. Quantification of Volumetric Oil Deposition

Unenhanced abdominal CT scans were performed 24 h–72 h before each cTACE procedure with a multislice CT scanner (Discovery CT750 HD (GE Medical System), Sensation 64 CT (Siemens), Somatom Definition (Siemens)). Standard liver scan protocol was used in the present study and can be seen in our recent research [17].

In the training cohort, images were interpreted independently by two radiologists (who had experience interpreting liver images for at least 5 years). In the validation cohort, images were interpreted by a single radiologist (who had experience in interpreting liver images for at least 5 years). Overall tumor volumes, as well as the amount of lipiodol deposition (in cm^3^), were measured using semiautomated quantification software (ITK-SNAP software (http://www.itksnap.org/pmwiki/pmwiki.php)). The total tumor volume was measured using pretreatment portal venous phase imaging, while the volume of lipiodol deposition was determined using noncontrast imaging. The lipiodol deposition rate was recorded as the ratio of the oil deposition volume to the total tumor volume. The rate of lipiodol retention before each TACE was classified as follows: (1) high level: >50% tumor volume and categorized as a responder; and (2) low level: ≤50% tumor volume and categorized as a nonresponder (Figure 2). The index lesion method was used to determine the tumor response [19,20,21]. The related clinical data were extracted from the electronic medical record system at each institution. The characteristics of the tumors, including the largest tumor size, number of lesions (with either three or more tumors, regardless of size) [22], and tumor capsule, were determined by a radiologist with experience in liver imaging.

### 2.4. Statistical Analysis

The clinical data and imaging characteristics were assessed by Student’s t-test, the chi-squared test, or the Mann–Whitney U test, as appropriate. Interobserver reproducibility was assessed using the intraclass correlation coefficient (ICC).

The Kaplan–Meier (KM) method was used for the survival curves and was compared with the results of the log-rank test. Univariate and multivariate Cox regression analyses were conducted to identify the independent predictive factors, and a nomogram was built. For example, nomogram I represented the pre-TACE-2 lipiodol deposition combined with pretreatment valuable predictors, and nomogram II represented the pre-TACE-3 lipiodol deposition combined with pretreatment valuable predictors (Figure 3). The performance of each model was evaluated with the concordance index (C-index). The highest diagnostic nomogram was compared with eight well-recognized models (six-and-twelve, seven-eleven criteria, hepatoma arterial embolization prognostic (HAP) score, modified HAP 3 score, BCLC-B subclassification system, ALBI grade, and assessment for retreatment (ART); as well as the alpha fetoprotein, Barcelona Clinic Liver Cancer, Child–Pugh increase, and tumor response (ABCR) score) by the time-dependent area under receiver-operating characteristic curve (AUROC). The clinical utility of the nomogram in both datasets was evaluated by calibration curve and a decision curve analysis (DCA).

In addition, the training set was divided into two subgroups (high- and low-score groups) based on the best cutoff points obtained from the “surv_cutpoint” function of the “survminer” R package. Patients in the validation sets were also categorized into two subgroups based on the same best cutoff points used in the training set. OS rates at 1, 2, and 3 years were calculated for each group, and KM survival curves were generated. Statistical significance was defined as *p* < 0.05. All statistical analyses were performed using R software (version 3.6.2).

## 3. Results

### 3.1. Baseline Characteristics

A total of 510 patients with HCC received TACE as monotherapy between May 2010 and May 2017. A total of 142 patients were included in the training cohort, and 87 were enrolled in the validation cohort. The median follow-up time was 45.9 months (interquartile range, 23.3–58.2 months) for the training cohort and 42.4 months (interquartile range, 27.2–62.2 months) for the validation cohort. The median OS values of the training and validation cohorts were not significantly different (median OS 31.4 [95% CI 23.2–36.2] months vs. 29.6 [95% CI 23.7–37.1] months, *p* = 0.212). The median number of TACE sessions for both cohorts was five (range, 4–7). Table 1 describes the baseline characteristics of patients in these cohorts before the first cTACE treatment.

### 3.2. Interobserver Agreement

Lipiodol retention levels were assessed by the ICC test, and excellent interobserver reproducibility was demonstrated (ICC, 0.866; 95% CI, 0.812–0.914).

### 3.3. Consecutive Oil Deposition

For the training cohort, the response rate to the first cTACE was 76 of 142 (53.5%; 95% CI: 43.7, 67.7). Among patients who did not respond to the first cTACE, the response rate after the second cTACE was 34 of 66 (51.5%; 95% CI: 47.8, 54.6). Among patients who did not respond to second cTACE, the response rate after the third cTACE was 7 of 32 (21.8%; 95% CI: 18.7, 23.7). Subsequently, among patients who did not respond to third cTACE, the corresponding rate after the fourth cTACE was low, at 3 of 25 (12%; 95% CI: 9.1, 15.2) (Table 2).

In the training cohort, the overall survival rates at 1, 2, and 3 years were significantly lower for nonresponders to the first cTACE than responders (*p* < 0.001). For patients who did not respond to the first cTACE but who underwent the second cTACE, overall survival rates at 1, 2, and 3 years after a the second cTACE were significantly lower for nonresponders than for responders (*p* < 0.001). For patients who did not respond to the second cTACE but received a third cTACE, long-term survival outcomes were similar for nonresponders and responders (*p* = 0.198). A similar trend was observed in patients who did not respond to the third cTACE but received the fourth cTACE (*p* = 0.268) (Table 3).

### 3.4. Model Building and Evaluation

Based on the results of the Cox proportional hazards model, the largest tumor size (all *p* < 0.001 for nomograms I, II, and III), the tumor number (all *p* < 0.001 for nomograms I, II, and III), and the pretreatment lipiodol deposition (*p* = 0.032 and *p* < 0.001 for nomograms I and II, respectively) were identified as independent predictors of survival for the training cohort. Several prediction models (nomograms I, II, and III) were constructed to predict patient prognosis (Table 4, Appendix A). Nomogram II had a higher predictive value than nomograms I and III (Table 5) and is shown in Figure 4. In comparison to eight other well-recognized prognostic models, the time-dependent AUROC analysis showed that nomogram II had improved diagnostic performance in the training cohort (Table 6; Figure 5A).

### 3.5. Validation

After the new prognostic model (nomogram II) was established, its performance was evaluated. In the validation cohort, nomogram II maintained adequate discriminative performance (C-statistic of 0.71; 95% CI, 0.68–0.73). Generally, the time-dependent AUROC values of nomogram II were higher than those of the other models in the validation cohort (Figure 5B). In addition, the calibration curves demonstrated the favorable calibration of nomogram II in both cohorts (Figure 6A,B). The decision curves displayed a good performance of nomogram II in terms of clinical utility (Figure 6C).

### 3.6. Survival Stratification

To identify the prognostic stratification of patients receiving cTACE, we calculated the optimal cutoff points for nomogram II in the training set using the “surv_cutpoint” function of the “survminer” R package, which was 70 points. Therefore, patients in the training cohort were divided into two subgroups: the low-score group (total score ≤ 70) and the high-score group (total score > 70). The KM survival analysis showed that the OS in the training group was significantly different between the two subgroups (median OS values of the low-score group and the high-score group were 42.1 months (95% CI 37.2–48.2) and 23.4 months (95% CI 17.2–28.2), respectively; log-rank test: *p* < 0.05) (Figure 7A). A similar trend was observed in the validation set (median OS values of the low-score group and high-score group were 43.4 months (95% CI 37.5–49.1) and 21.2 months (95% CI 16.9–26.3), respectively; log-rank test: *p* < 0.05) (Figure 7B).

## 4. Discussion 

The majority of HCC patients who undergo TACE have varying degrees of cirrhosis, which may limit the potential survival benefit of the procedure. This is further complicated by the fact that HCC patients commonly need multiple TACE sessions to obtain an optimal tumor response [23]. Hiraoka et al. [8] suggested that 9–14% of liver function deteriorates to CP-B after each TACE procedure, and hepatic function is also essential in the tyrosine kinase inhibitor (TKI) and immunotherapy eras. Therefore, evaluation of the benefits of each TACE is critical to decision-making in clinical practice. Our study had two major findings. First, we confirmed that approximately 50% of patients who were nonresponsive to the first cTACE showed a significant response after the second TACE, which is consistent with results by Georgiades et al. [24]. Second, the nomogram constructed by pre-TACE-3 lipiodol deposition combined with clinical variables could be used to predict relatively long-term patient survival, and its diagnostic value was higher than that of other existing clinical predictive models.

Lipiodol deposition has been proved to be strongly associated with OS in patients with BCLC B-stage HCC treated with cTACE [12]. Several studies have shown that lipiodol deposition on intraoperative CBCT can be utilized to predict tumor response [13,25]. However, CBCT has inherent shortcomings. CBCT clarity and real resolution are not equivalent to CT; moreover, motion artifacts, especially from respiration, can affect imaging analysis. As an alternative, magnetic resonance imaging (MRI) can be used to diagnose residual tumors after cTACE in a timely manner, with superior performance compared to CT [26]. However, peritumoral inflammation due to TACE can also lead to a false-positive diagnosis of tumor necrosis [27]. An inaccurate assessment of treatment response may have harmful consequences, especially misleading the interventional radiologist to subsequently choose an inappropriate TACE. Varzaneh et al. [28] reported that post-TACE lipiodol deposition in MDCT could accurately predict tumor necrosis (treatment response) in treated HCC lesions. Therefore, we used CT imaging to evaluate the tumor treatment response and found that the lipiodol deposition level combined with pretreatment clinical data could be used to predict HCC patients who underwent multiple TACEs.

It remains controversial whether treatment should be changed during repeated TACE or whether the effect obtained at a certain time helps to predict patient survival [29]. The optimal number of sessions before abandoning cTACE or requiring combined treatment is also controversial. The Japan Society of Hepatology (JSH) proposed “TACE refractoriness” and recommended that at least two TACE treatments be performed before abandonment [30]. This TACE refractoriness concept has gradually been accepted by various HCC panels [22,31,32]. Notably, in the present study, after the second TACE procedure, approximately 50% of patients with BCLC B-stage HCC who did not respond to the first chemoembolization procedure showed a significant response, while less than 25% of patients who did not respond to the second and third cTACE sessions responded to the third and fourth sessions, respectively. Therefore, two sessions of TACE were sufficient to evaluate the treatment response and, thereafter, patients could consider abandoning cTACE or the need for combined treatment. This result was consistent with the recommendation of the European Association for the Study of the Liver (EASL) guideline that cTACE should be abandoned when a significant tumor treatment response has not been achieved after two cycles of treatment [33]. Regarding survival analyses, the results showed that a largest tumor size greater than 5 cm and a tumor number over 3 were independently associated with inferior OS; these results are consistent with previous studies [34,35].

In recent years, several individualized prediction models have been established to evaluate patient prognosis after cTACE, including tumor burden evaluation models such as the six-and-twelve system and liver reserve evaluation systems such as ALBI grade. Our present study proved that the nomogram based on pre-TACE-3 lipiodol deposition had the highest value for predicting patient outcomes among all the analyzed models. Compared with our constructed nomogram, current commonly used prediction models were built by preprocedure clinical variables. The ALBI grade can provide an objective method for assessing liver function in HCC patients with good prognosis [36]. However, a study by Chi et al. [37] demonstrated that “ALBI-grade migration” was an independent risk factor associated with poor progress-free survival (PFS) and short-term OS. Research by Hiraoka et al. [8] and Lin et al. [38] illustrated that dynamic changes in the ALBI score are a good predictive parameter for prognosis in patients receiving cTACE. Therefore, the application of pretreatment prediction (which cannot independently assess the effect of predictor changes on outcome) and posttreatment prediction models (which cannot independently evaluate the effect of a patient’s underlying condition on outcomes) alone may not be sufficient to accurately predict outcome. Recent studies by Adhoute et al. [39] and Wang et al. [40] concluded that pretreatment clinical variables (such as tumor size and tumor number) combined with post-TACE data (presence or absence of radiological response) could have higher values than other pretreatment prediction and posttreatment prediction models. The newly built nomogram combined the best predictive variables in the pretreatment and posttreatment periods to achieve the optimum prediction, which, not surprisingly, was more efficient than other well-known prediction models. The recent 2022 BCLC guidelines were updated, and two novel concepts were introduced: treatment stage migration (TSM) and untreatable progression [41]. Untreatable progression is defined when either treatment failure or progression after the selected treatment approach occur, but patients still fit into their initial BCLC stage, thus warranting the consideration of a therapy corresponding to a more advanced stage. Given the Barcelona Clinic Liver Cancer 2022 update and the successful validation of our newly constructed model (nomogram II), we proposed a novel algorithm for TACE retreatment (Figure 8). 

To guide the best options for TACE retreatment in HCC patients, Sieghart et al. [42] developed a scoring system called the ART score. Another published scoring system, known as the ABCR score, also aims to select patients who are not capable of benefitting from continued TACE [43]. A time-dependent AUC curve analysis showed that the prognostic value was significantly lower than that of our newly constructed model. There are two major reasons to explain the results: (1) In our study, as well as in research by Georgiades et al. [24], it was demonstrated that, after one TACE session, evaluating the tumor response may not be appropriate. (2) The ART and ABCR scores use the EASL criteria to assess tumor response; however, in tumors with patchy and irregular necrosis, compared to computer-assisted semiautomated measurement, the accurate determination of the tumor border is problematic and susceptible to interobserver variation when using the EASL criteria [10].

The present study has some limitations. First, this retrospective study had potential patient selection bias. Second, the sample size was small; therefore, our results need to be validated with a larger sample size. Finally, the accuracy of semiautomated volume measurement is limited by computer technologies and is dependent on precise contour delineation. Radiomics, a burgeoning technology that could transform potential pathological and physiological information from routine-acquired images into high-dimensional, quantitative, and mineable imaging data, has been demonstrating great potential in the survival predictions of HCC [17,44]. In the future, we aim to build a larger database and use artificial intelligence (AI) for data processing.

## 5. Conclusions

In summary, based on pre-TACE lipiodol deposition, two sessions were recommended before abandoning cTACE or considering the need for combined treatment for patients with intermediate-stage HCC. Furthermore, we developed and validated a relatively reliable prognostic nomogram to predict the long-term OS in patients with BCLC B-stage HCC after cTACE. Subsequently, the use of this nomogram should be encouraged to improve decision making by providing individualized survival information.

## Figures and Tables

**Figure 1 jpm-12-01375-f001:**
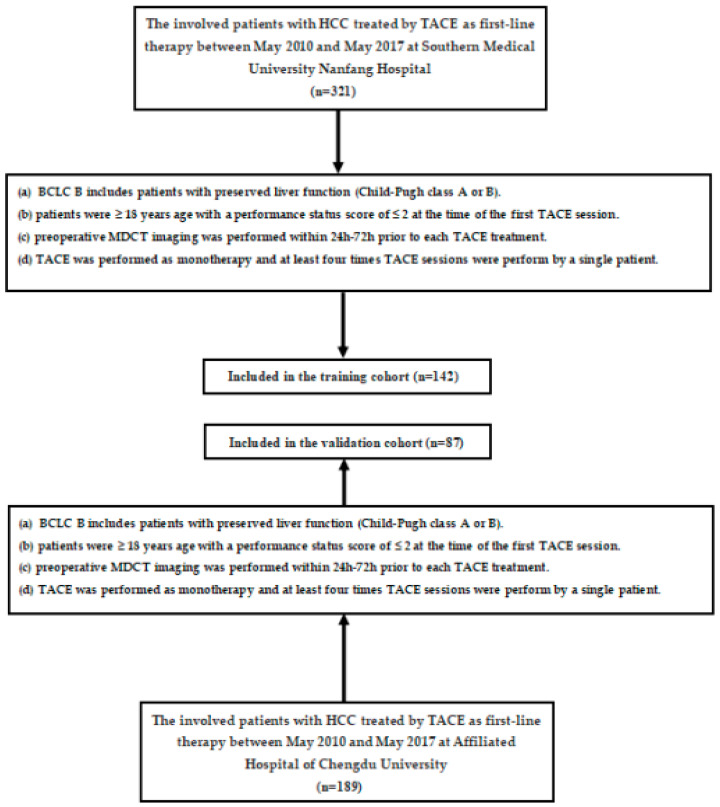
Flowchart for inclusion and exclusion of patients within the training and validation cohorts. TACE: transarterial chemoembolization; BCLC: Barcelona Clinic Liver Cancer; MDCT: multi-detector computed tomography; HCC: hepatocellular carcinoma.

**Figure 2 jpm-12-01375-f002:**
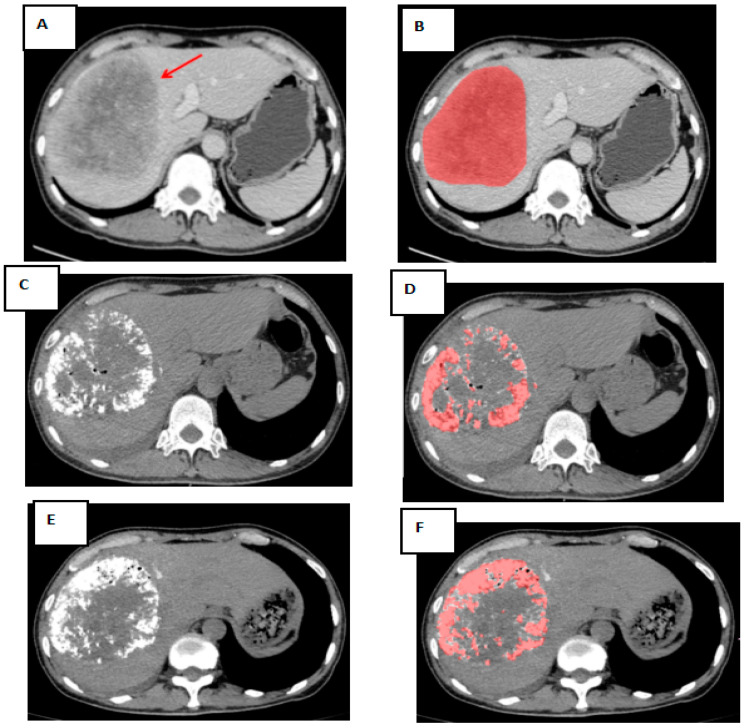
3D volumetric semi-automatic evaluation of diffuse lipiodol retention from one representative patient. (**A**): A large HCC in right hepatic lobe (red arrow) seen on preprocedural contrast-enhanced CT images as a hypoattenuating tumor in the portal venous phase. (**B**): The red shaded area depicts semi-automated segmentation of the tumor; the tumor volume on the pretreatment MDCT was 512.6 cm^3^. (**C**): Lipiodol retention on noncontrast CT imaging in pre-TACE-2. (**D**): The red shaded area depicts the lipiodol volume on noncontrast CT imaging of 121.7 cm^3^ in pre-TACE-2. (**E**): Lipiodol retention on noncontrast CT imaging in pre-TACE-3. (**F**): The red shaded area demonstrates the lipiodol volume on noncontrast CT imaging as 174.8 cm^3^ in pre-TACE-3. After two cycles of TACEs, the volume of lipiodol was less than 50% of total tumor volume. TACE: transarterial chemoembolization; BCLC: Barcelona Clinic Liver Cancer; MDCT: multi-detector computed tomography; HCC: hepatocellular carcinoma.

**Figure 3 jpm-12-01375-f003:**
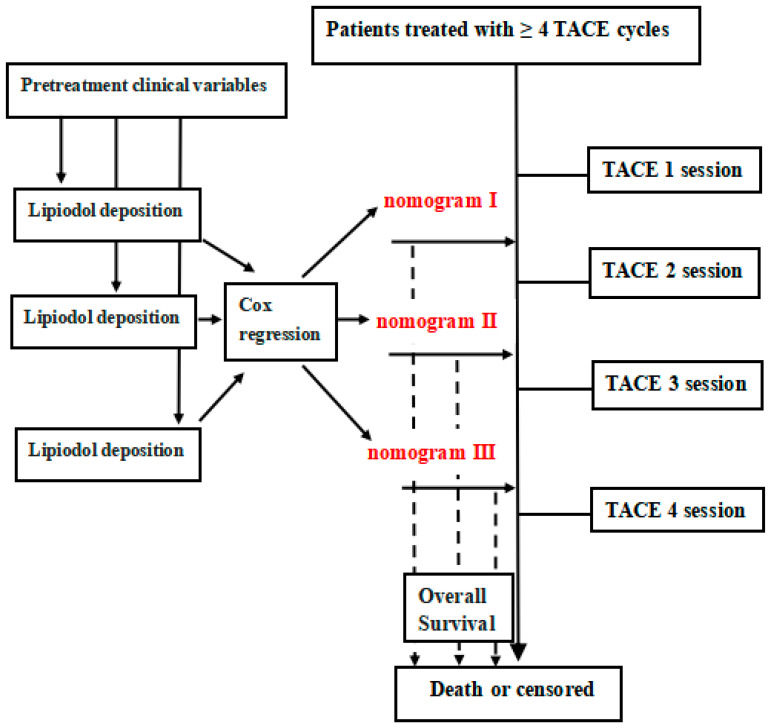
Study design for nomogram construction. Nomograms were built by Cox regression analyses based on each pretreatment lipiodol deposition and clinical variables. TACE: transarterial chemoembolization.

**Figure 4 jpm-12-01375-f004:**
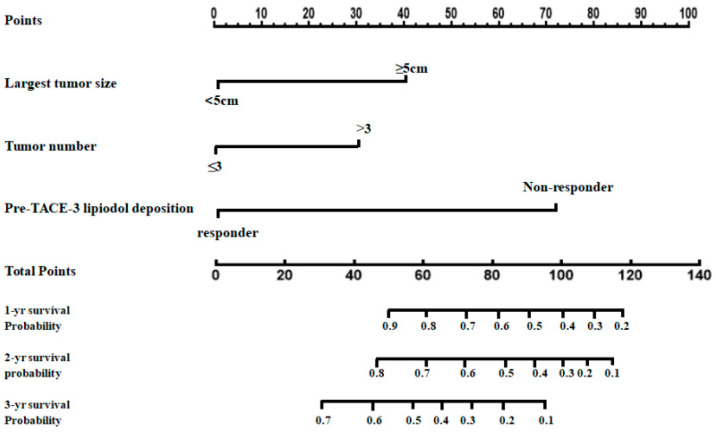
Prognostic nomogram showing the assessment of 1-, 2-, and 3-year survival in patients with intermediate-stage HCC after cTACE. (To use this nomogram in clinical practice, a vertical line is first drawn from the factor axis to the “Points” scale to determine the number of points for each factor (“Largest tumor size,” “Tumor number,” and “Pre-TACE-3 lipiodol deposition.”) Then, these numbers are summed and located on the axis of the total points. Finally, a downward line is drawn from the axis of the total points to the survival axes to calculate the 1-, 2-, and 3-year survival probabilities.) TACE: transarterial chemoembolization.

**Figure 5 jpm-12-01375-f005:**
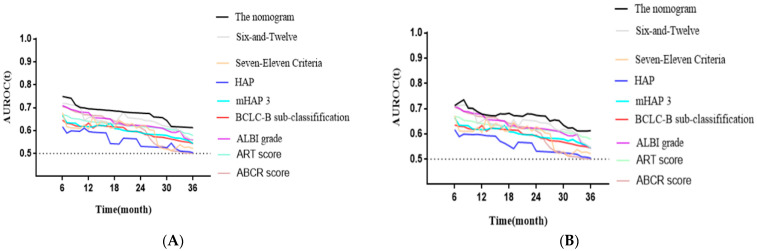
Time-dependent AUROC values of the current model and other available models. (**A**): Time-dependent AUROC values in training set. (**B**): Time-dependent AUROC values in validation set. HAP: hepatoma arterial embolization prognostic; mHAP 3: modified HAP 3; ALBI: albumin-bilirubin; ART: assessment for retreatment; ABCR: alpha fetoprotein, Barcelona Clinic Liver Cancer, Child–Pugh increase, tumor response; AUROC: area under receiver-operating characteristic curve; BCLC: Barcelona Clinic Liver Cancer.

**Figure 6 jpm-12-01375-f006:**
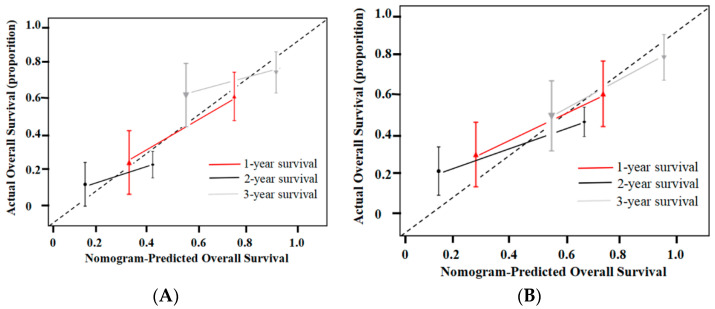
Calibration and decision curve analyses. (**A**): Calibration curves of the nomogram on the training dataset. The Hosmer–Lemeshow test yielded a *p*-value of 0.121 in the training dataset. (**B**): Calibration curves of the nomogram on the validation dataset. The Hosmer–Lemeshow test yielded a *p*-value of 0.137 in the validation dataset. (**C**): Decision curve analysis for the newly constructed nomogram (nomogram II) in the training cohort and the validation cohort. The *y*-axis represents the net benefit, and the *x*-axis represents the threshold probability. The newly constructed nomogram obtained more benefit than either the treat-all-patients scheme (gray line) or the treat-none scheme (horizontal black dashed line) within certain ranges of threshold probabilities for predicting therapeutic response to sequential cTACE in HCC.

**Figure 7 jpm-12-01375-f007:**
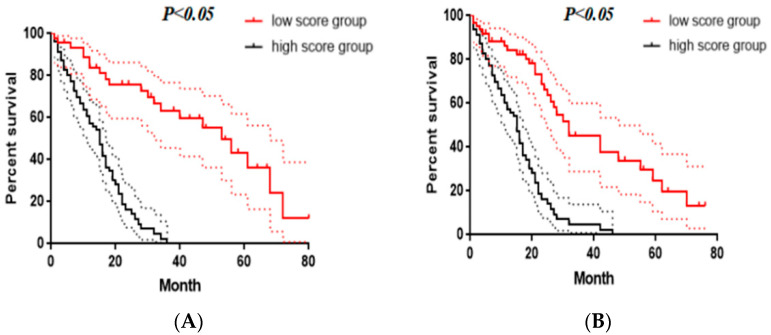
Kaplan–Meier survival curves for overall survival (OS) of patients according to the nomogram-II-based subgroups in the training set (**A**) and validation set (**B**).

**Figure 8 jpm-12-01375-f008:**
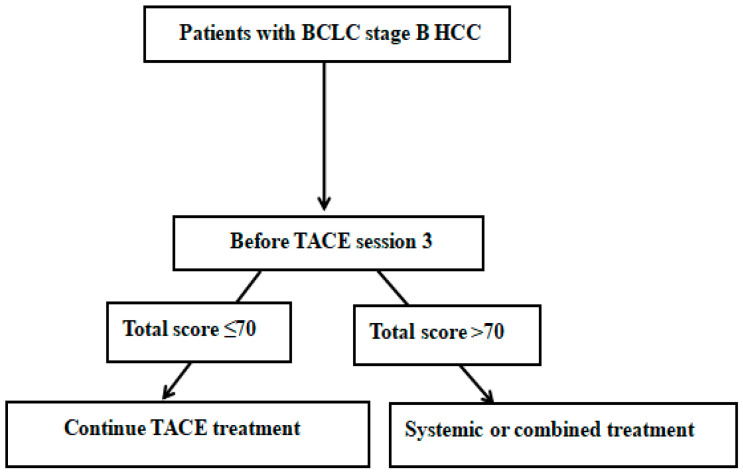
Proposal for nomogram-II-guided retreatment strategy with TACE. TACE: transarterial chemoembolization; BCLC: Barcelona Clinic Liver Cancer; HCC: hepatocellular carcinoma.

**Table 1 jpm-12-01375-t001:** Patient characteristics.

Characteristic	Training Cohort		Validation Cohort		*p*-Value
	*N =* 142	%	*N =* 87	%	
**Age (yr)**					0.731
<60	90	63	52	60	
≥60	52	37	35	40	
**Sex**					0.329
Male	95	67	49	56	
Female	47	33	38	44	
**HBsAg status**					0.941
Positive	112	79	69	80	
Negative	30	21	18	20	
**Child–Pugh class**					0.881
A	89	62	54	62	
B	53	38	33	38	
**Largest tumor size (cm)**					0.227
<5 mean ± SD	96	67	47	54	
≥5	46	33	40	46	
**Tumor number**					0.258
≤3	91	64	49	56	
>3	51	36	38	44	
**AFP ((IU/mL)**					0.341
<200	40	28	27	31	
≥200	102	72	60	69	
**AST (U/L)**					0.319
<40	50	35	37	42	
≥40	92	65	50	58	
**ALT(U/L)**					0.431
<40	53	37	31	35	
≥40	89	63	56	65	
**ALB (g/L)**					0.351
<35	41	29	21	24	
≥35	101	71	66	76	
**Capsule**					0.239
Absent	67	47	32	36	
Present	75	53	55	64	
**Up-to-seven criteria**					
Within	44	30	22	25	0.651
Beyond	98	70	65	75	

BCLC: Barcelona Clinic Liver Cancer; HBsAg: hepatitis B surface antigen; AST: aspartate aminotransferase; ALT: alanine transaminase; AFP: alpha fetoprotein; ALB: albumin.

**Table 2 jpm-12-01375-t002:** Nonresponders to previous cTACEs after next cTACE sessions in the training cohort.

Parameter	No. of Patients	Responders	Nonresponders
** *Response after Second Chemoembolization* **			
** *Response after First* ** ** *Chemoembolization* **			
Nonresponders	66	34 (51.5) (47.8, 54.6)	32 (48.5) (42.2, 52.3)
** *Response after Third Chemoembolization* **			
** *Response after Second* ** ** *Chemoembolization* **			
Nonresponders	32	7 (21.8) (18.7, 23.7)	25 (78.2) (71.3, 82.4)
** *Response after Third* ** ** *Chemoembolization* **			
** *Response after Fourth Chemoembolization* **			
Nonresponders	25	3 (12.0) (9.1, 15.2)	22 (88.0) (82.9, 91.4)

Data in parentheses are percentages; data in brackets are 95% CIs.

**Table 3 jpm-12-01375-t003:** Survival outcomes for responders and nonresponders in the training cohort.

Parameter	Median OS (mo)	1-Year OS	2-Year OS	3-Year OS	*p*-Value
**Survival outcomes after cTACE-1**					*p* < 0.001
Responders	39.2 ± 0.5	91.4 (82.6, 92.9)	56.2 (47.2, 60.4)	33.8 (28.2, 38.2)	
Nonresponders	26.1 ± 1.2	88.4 (81.2, 91.3)	26.1 (21.1, 30.2)	13.4 (10.1, 18.8)	
**Survival outcomes for nonresponders to cTACE-1 who underwent cTACE-2**					*p* < 0.001
Responders	38.1 ± 0.9	90.2 (85.2, 94.1)	60.1 (56.2, 68.1)	36.2 (32.2, 40.1)	
Nonresponders	24.2 ± 0.8	86.2 (81.3, 90.5)	28.9 (22.1, 31.3)	14.8 (11.2, 18.2)	
**Survival outcomes for nonresponders to cTACE-2 who underwent cTACE-3**					*p* = 0.198
Responders	31.2 ± 0.7	86.4 (81.2, 91.3)	57.1 (48.1, 62.2)	30.7 (26.4, 35.2)	
Nonresponders	28.4 ± 0.9	85.2 (82.7, 90.2)	54.3 (48.1, 61.2)	26.5 (22.1, 30.4)	
**Survival outcomes for nonresponders to cTACE-3 who underwent cTACE-4**					*p* = 0.268
Responders	19.1 ± 0.9	61.4 (55.2, 66.3)	14.2 (10.9, 18.4)	8.2 (5.2, 11.7)	
Nonresponders	17.4 ± 1.6	59.2 (54.9, 65.7)	10.8 (6.2, 14.3)	6.1 (2.2, 10.8)	

Note: median data are means ± standard deviation; data in parentheses are 95% CIs. cTACE-1: first conventional transarterial chemoembolization (cTACE) session; cTACE-2: second cTACE session; cTACE-3: third cTACE session; cTACE-4: fourth cTACE session; OS: overall survival.

**Table 4 jpm-12-01375-t004:** Predictors of death for patients with intermediate-stage HCC before second TACE.

Variable	Univariate Analysis			Multivariate Analysis		
Before Second TACE	Hazard Ratio	95% CI	*p*-Value	Hazard Ratio	95% CI	*p*-Value
**Age (yr)** <60/≥60	0.68	0.32–1.03	0.171			
**Sex** male/female	0.81	0.42–1.27	0.233			
**HBsAg status** positive/negative	0.54	0.23–0.76	0.234			
**Child–Pugh class** A/B	1.32	0.63–1.34	0.338			
**Largest tumor size (cm)** <5/≥5	1.34	1.17–3.04	0.007	1.56	0.45–2.43	<0.001
**Tumor number** ≤3/m>3	1.31	1.01–2.98	0.004	1.45	1.07–3.00	<0.001
**AFP (IU/mL)** <200/≥200	0.62	0.22–1.09	0.018	1.45	1.04–2.01	0.078
**AST (U/L)** <40/≥40	0.54	0.27–0.94	0.081			
**ALT(U/L)** <40/≥40	0.53	0.12–0.78	0.079			
**ALB (g/L)** <35/≥35	1.34	0.82–2.34	0.043	1.45	1.01–1.71	0.073
**Capsule** absent/present	1.87	1.34–3.97	0.018	1.46	0.09–2.76	0.093
**Up-to-seven criteria** within/beyond	1.32	1.10–1.87	0.032	1.67	1.00–1.76	0.098
**Pre-TACE-2 lipiodol deposition** **responder/nonreponder**	1.89	0.71–2.87	0.038	1.98	0.87–2.89	0.032

BCLC: Barcelona Clinic Liver Cancer; HBsAg: hepatitis B surface antigen; AST: aspartate aminotransferase; ALT: alanine transaminase; AFP: alpha fetoprotein; ALB: albumin.

**Table 5 jpm-12-01375-t005:** Comparison of the performance and discriminative ability of constructed nomogram models.

Cohort	Models	1-yr AUROC (95% CI)	2-yr AUROC (95% CI)	3-yr AUROC (95% CI)	C-Index (95% CI)	*p*-Value
**Training**	Nomogram I	0.70 (0.69–0.73)	0.65 (0.63–0.67)	0.61 (0.60–0.63)	0.66 (0.64–0.68)	0.032
	Nomogram II	0.74 (0.71–0.77)	0.71 (0.69–0.73)	0.67 (0.65–0.69)	0.72 (0.69–0.74)	Ref.
	Nomogram III	0.61 (0.59–0.66)	0.59 (0.53–0.63)	0.53 (0.50–0.57)	0.56 (0.52–0.62)	<0.01
**Validation**	Nomogram I	0.67 (0.65–0.69)	0.64 (0.61–0.67)	0.61 (0.59–0.63)	0.63 (0.60–0.65)	0.028
	Nomogram II	0.72 (0.69–0.74)	0.70 (0.68–0.73)	0.65 (0.63–0.67)	0.71 (0.68–0.73)	Ref.
	Nomogram III	0.59 (0.53–0.63)	0.57 (0.53–0.61)	0.52 (0.50–0.57)	0.53 (0.51–0.57)	<0.01

AUROC: area under receiver-operating characteristic curve; CI: confidence interval; Ref.: reference.

**Table 6 jpm-12-01375-t006:** Comparison of the performance and discriminative ability between the current model and other models.

Cohort	Models	1-yr AUROC (95% CI)	2-yr AUROC (95% CI)	3-yr AUROC (95% CI)	C-Index (95% CI)	*p*-Value
**Training**	Nomogram II	0.74 (0.71–0.77)	0.71 (0.69–0.73)	0.67 (0.65–0.69)	0.72 (0.69–0.74)	Ref.
	Six-and-Twelve	0.71 (0.69–0.73)	0.66 (0.64–0.68)	0.62 (0.60–0.65)	0.67 (0.65–0.69)	0.032
	Seven-Eleven Criteria	0.69 (0.67–0.71)	0.63 (0.61–0.67)	0.59 (0.55–0.62)	0.63 (0.61–0.65)	0.023
	HAP	0.55 (0.51–0.59)	0.54 (0.52–0.57)	0.53 (0.50–0.55)	0.56 (0.53–0.59)	<0.01
	mHAP 3	0.61 (0.59–0.63)	0.59 (0.54–0.62)	0.52 (0.50–0.55)	0.61 (0.58–0.64)	<0.01
	BCLC-B subclassification	0.58 (0.56–0.63)	0.54 (0.52–0.58)	0.52 (0.50–0.56)	0.57 (0.55–0.61)	<0.01
	ALBI grade	0.66 (0.64–0.69)	0.61 (0.59–0.64)	0.59 (0.55–0.65)	0.63 (0.60–0.69)	0.038
	ART score	0.61 (0.58–0.64)	0.59 (0.56–0.63)	0.57 (0.54–0.60)	0.60 (0.56–0.64)	<0.01
	ABCR score	0.67 (0.63–0.70)	0.65 (0.62–0.68)	0.61 (0.57–0.65)	0.65 (0.62–0.69)	0.036
**Validation**	Nomogram II	0.72 (0.69–0.74)	0.70 (0.68–0.73)	0.65 (0.63–0.67)	0.71 (0.68–0.73)	Ref.
	Six-and-Twelve	0.70 (0.68–0.73)	0.65 (0.62–0.68)	0.61 (0.58–0.63)	0.66 (0.64–0.69)	0.048
	Seven-Eleven Criteria	0.68 (0.66–0.71)	0.62 (0.59–0.65)	0.57 (0.55–0.61)	0.62 (0.60–0.64)	0.041
	HAP	0.54 (0.51–0.56)	0.52 (0.50–0.55)	0.50 (0.49–0.53)	0.54 (0.52–0.56)	<0.01
	mHAP 3	0.60 (0.57–0.63)	0.56 (0.52–0.61)	0.51 (0.48–0.54)	0.60 (0.58–0.63)	<0.01
	BCLC-B subclassification	0.56 (0.54–0.60)	0.52 (0.50–0.55)	0.50 (0.48–0.53)	0.55 (0.53–0.60)	<0.01
	ALBI grade	0.65 (0.62–0.67)	0.60 (0.58–0.63)	0.56 (0.54–0.60)	0.61 (0.58–0.64)	0.039
	ART score	0.62 (0.57–0.66)	0.56 (0.52–0.60)	0.54 (0.51–0.58)	0.59 (0.56–0.63)	<0.01
	ABCR score	0.68 (0.64–0.71)	0.66 (0.62–0.70)	0.63 (0.59–0.67)	0.66 (0.62–0.70)	0.041

AUROC: area under receiver-operating characteristic curve; CI: confidence interval; HAP: hepatoma arterial embolization prognostic; mHAP: modified HAP; BCLC: Barcelona Clinic Liver Cancer; ALBI: albumin-bilirubin; ART: assessment for retreatment; ABCR: alpha fetoprotein, Barcelona Clinic Liver Cancer, Child–Pugh increase, tumor response; Ref.: reference.

## Data Availability

The original contributions presented in the study are included in the article and Appendix A; further inquiries can be directed to the corresponding author.

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
