# Peer review of "A Nomogram Based on Preoperative Lipiodol Deposition after Sequential Retreatment with Transarterial Chemoembolization to Predict Prognoses for Intermediate-Stage Hepatocellular Carcinoma"

_jpm, 2022, doi:10.3390/jpm12091375_

Round 1

Reviewer 1 Report

Overall the manuscript is well written.

Manuscript findings introduce new questions and can represent the basis of future studies on the topic.

Methodology of the study is clear, as well as the results and the conclusions.

Figures and Tables are clear and well described. 

However, I noticed some refuses, and have as well a few suggestions to improve the manuscript:

 - Please revise the punctuation and text spacing throughout the text.

- The "p" in "p-value" should be always written in lowercase italics letters; please change this throughout the manuscript.

- Figure 2: please reduce the lenght of figure legend text.

 - References are appropriate; however, there is need to format all the references according to the Journal's standards.

- References: mention of the 2022 updated BCLC guidelines should be made (doi: 10.1016/j.jhep.2021.11.018.); mention of predictive value of radiomic analysis after locoregional treatments for HCC, even if still in its infancy, should be made in the discussion (doi: 10.26355/eurrev_202204_28620 and other similar articles)

Author Response

Comments and Suggestions for Authors

Overall the manuscript is well written.

Manuscript findings introduce new questions and can represent the basis of future studies on the topic.

Methodology of the study is clear, as well as the results and the conclusions.

Figures and Tables are clear and well described.

Thank you for your precious comments. It is very helpful to improve my article. And thank JPM gives me the opportunity to revise my research. According to your suggestions, we have revised it. Hopefully it can reach

JPM requirement. 

 However, I noticed some refuses, and have as well a few suggestions to improve the manuscript: 

 - Please revise the punctuation and text spacing throughout the text.

Response to Rev. 1, comment 1.

We thank the reviewer for this important comment. We have revised the punctuation and text spacing throughout the paper, please check it.

- The "p" in "p-value" should be always written in lowercase italics letters; please change this throughout the manuscript.

Response to Rev. 1, comment 2.

We thank the reviewer for this important comment. We have changed it throughout the manuscript, please check it.

- Figure 2: please reduce the lenght of figure legend text.

Response to Rev. 1, comment 3.

We thank the reviewer for this important comment. We have reduced the the lenght of figure legend text in Fig. 2, please check it.

 - References are appropriate; however, there is need to format all the references according to the Journal's standards.

Response to Rev. 1, comment 3.

We thank the reviewer for this important comment. We have modified the  reference format according to the requirements of JMP.

- References: mention of the 2022 updated BCLC guidelines should be made (doi: 10.1016/j.jhep.2021.11.018.); mention of predictive value of radiomic analysis after locoregional treatments for HCC, even if still in its infancy, should be made in the discussion (doi: 10.26355/eurrev_202204_28620 and other similar articles)

Response to Rev. 1, comment 4.

We thank the reviewer for this important comment. We have mentioned the the 2022 updated BCLC guidelines, please check reference 41. We also added the article (doi: 10.26355/eurrev_202204_28620) in the limitation section (please check reference 45) and given appropriate discussion. modified the  reference format according to the requirements of JPM.

Reviewer 2 Report

Give a statement regarding multiple small lesions and a picture as example.Please give the right year regarding literature number 42 - is it 2001?

HAZARD RATIO: give only two numbers after the point e.g 1.23 and not 1.345

Author Response

Comments and Suggestions for Authors

Thank you for your precious comments. It is very helpful to improve my article. And thank JPM gives me the opportunity to revise my research. According to your suggestions, we have revised it. Hopefully it can reach JPM requirement.

Give a statement regarding multiple small lesions and a picture as example.

Response to Rev. 2, comment 1.

We thank the reviewer for this important comment. We have explained the multiple lesions, and already given the appropriate reference, please check ref. 22.

Please give the right year regarding literature number 42 - is it 2001?

Response to Rev. 2, comment 2.

We thank the reviewer for this important comment. We have modified the literature number 42, please check ref. 42.

HAZARD RATIO: give only two numbers after the point e.g 1.23 and not 1.345

Response to Rev. 2, comment 3.

We thank the reviewer for this important comment. We have modified according to suggestion, please check the table 4 and Table S1-S2.

Reviewer 3 Report

The authors present a paper about "A nomogram based on preoperative lipiodol deposition after sequential retreatment with transarterial chemoembolization to predict prognoses for intermediate-stage hepatocellular carcinoma".

The topic is interesting and the amount of patients included is sufficient.

The statistical is adequate and the methodology followed is clearly presented by the authors.

The resuts found are interesting and the conclusions are balanced and supported by the results.

However I have a major concern about the Informed consent statement: in fact the fact that it is a retrospcetive study does not authorize the authors to waive informed consent that sgiuld always be obtained for publicaation of personal data of patients. I suggest that the authors obtain and provide patients consent for publicatin of their personal data.

Round 2

Reviewer 3 Report

I have no further comments

Author Response

Comments and Suggestions for Authors

Editor' comments: I am however bothered by the absence of a sentence on patient consent and on the ethics of your research. I would also like to know if the scan performed 48-72 h before chemoembolisation is standard practice or research. We do it a month before in my centre. Could you please clarify theses two points and add a few lines in the method section (ethical aspect) before publication?

Thank you for your precious comments. In the present study, the CT scan performed 48-72 h before chemoembolisation is standarded practice in our centre. And the patients signed an informed consent before the CT scan and also before receiving TACE treatment. I have added a few lines in the method section to address the issue, please check it.
